# Signatures of Mitochondrial Dysfunction and Impaired Fatty Acid Metabolism in Plasma of Patients with Post-Acute Sequelae of COVID-19 (PASC)

**DOI:** 10.3390/metabo12111026

**Published:** 2022-10-26

**Authors:** Vamsi P. Guntur, Travis Nemkov, Esther de Boer, Michael P. Mohning, David Baraghoshi, Francesca I. Cendali, Inigo San-Millán, Irina Petrache, Angelo D’Alessandro

**Affiliations:** 1Division of Pulmonary and Critical Care and Sleep Medicine, Department of Medicine, National Jewish Health, Denver, CO 80206, USA; 2Division of Pulmonary Sciences and Critical Care Medicine, Department of Medicine, University of Colorado Anschutz Medical Campus, Aurora, CO 80045, USA; 3Department of Biochemical and Molecular Genetics, University of Colorado Anschutz Medical Campus, Aurora, CO 80045, USA; 4Department of Biostatistics, National Jewish Health, Denver, CO 80206, USA; 5Division of Endocrinology, Metabolism and Diabetes, Department of Medicine, University of Colorado Anschutz Medical Campus, Aurora, CO 80045, USA; 6Division of Medical Oncology, Department of Medicine, University of Colorado Anschutz Medical Campus, Aurora, CO 80045, USA; 7Department of Human Physiology and Nutrition, University of Colorado, Colorado Springs, CO 80918, USA

**Keywords:** keyword post-acute sequelae of COVID-19, long COVID, exercise intolerance, metabolomics, plasma

## Abstract

Exercise intolerance is a major manifestation of post-acute sequelae of severe acute respiratory syndrome coronavirus infection (PASC, or “long-COVID”). Exercise intolerance in PASC is associated with higher arterial blood lactate accumulation and lower fatty acid oxidation rates during graded exercise tests to volitional exertion, suggesting altered metabolism and mitochondrial dysfunction. It remains unclear whether the profound disturbances in metabolism that have been identified in plasma from patients suffering from acute coronavirus disease 2019 (COVID-19) are also present in PASC. To bridge this gap, individuals with a history of previous acute COVID-19 infection that did not require hospitalization were enrolled at National Jewish Health (Denver, CO, USA) and were grouped into those that developed PASC (n = 29) and those that fully recovered (n = 16). Plasma samples from the two groups were analyzed via mass spectrometry-based untargeted metabolomics and compared against plasma metabolic profiles of healthy control individuals (n = 30). Observational demographic and clinical data were retrospectively abstracted from the medical record. Compared to plasma of healthy controls or individuals who recovered from COVID-19, PASC plasma exhibited significantly higher free- and carnitine-conjugated mono-, poly-, and highly unsaturated fatty acids, accompanied by markedly lower levels of mono-, di- and tricarboxylates (pyruvate, lactate, citrate, succinate, and malate), polyamines (spermine) and taurine. Plasma from individuals who fully recovered from COVID-19 exhibited an intermediary metabolic phenotype, with milder disturbances in fatty acid metabolism and higher levels of spermine and taurine. Of note, depletion of tryptophan—a hallmark of disease severity in COVID-19—is not normalized in PASC patients, despite normalization of kynurenine levels—a tryptophan metabolite that predicts mortality in hospitalized COVID-19 patients. In conclusion, PASC plasma metabolites are indicative of altered fatty acid metabolism and dysfunctional mitochondria-dependent lipid catabolism. These metabolic profiles obtained at rest are consistent with previously reported mitochondrial dysfunction during exercise, and may pave the way for therapeutic intervention focused on restoring mitochondrial fat-burning capacity.

## 1. Introduction

Sometimes known as “post-COVID-19 syndrome” or “long COVID”, the syndrome of post-acute sequelae of SARS-CoV-2 infection (PASC) is a clinical condition characterized by lingering symptoms such as exercise intolerance for four weeks or more following acute COVID-19. PASC is associated with increased morbidity and a high societal cost, and the lack of available therapies [1] urge for a better understanding of its pathogenesis. We recently reported that, compared with historical pre-COVID era controls, individuals in a small cohort of PASC evaluated for exercise intolerance showed higher levels of arterial lactate and slower rates of fatty acid (FA) oxidation (FAtOx) during graded exercise testing, consistent with mitochondrial dysfunction [2,3]. Our findings, since corroborated since by others [4], provide the impetus for further insight into mitochondrial function and fatty acid metabolism in PASC.

To meet the demands of increasing load and duration of exercise requiring an accelerated capacity for adenosine triphosphate (ATP) production, mitochondria rely on metabolic flexibility to utilize various substrates. For physical efforts of extended duration (>1 min), ATP is preferentially generated via oxidative phosphorylation in the Tricarboxylic Acid Cycle (TCA). The substrates utilized in TCA to generate reducing equivalents that fuel the electron transport chain are FA (β-oxidation, FAtOx) and carbohydrates (CHOx). To enter the mitochondria, FA (acyl-CoAs) must be first conjugated by carnitine palmitoyl transferase (CPT) I to acylcarnitine, followed by the intra-mitochondrial exchange of carnitine for CoA by CPT II [5]. Lactate taken up from the cytosol is then oxidized in mitochondria by lactate dehydrogenase to pyruvate, which is transported across the inner mitochondrial membrane into the matrix, where pyruvate dehydrogenase generates acetyl-CoA for TCA metabolism and energy production [6]. The evidence of lower FAtOx in PASC may explain the inefficiency of mitochondria to sustain endurance exercise, whereas the premature lactate accumulation suggests either a metabolic shift in increased glycolysis or the inability to utilize lactate in the mitochondria as an alternative source of energy during exercise. A broader examination of metabolites in plasma may help to clarify metabolic derangements in PASC.

By measuring lactate and other small-molecule metabolites < 1.5 kDa, metabolomics offers a functional ‘post-genomic’ characterization of biochemical and signaling pathways influenced by COVID-19 [7,8,9,10,11,12], and provides noninvasive measurements of mitochondria-related metabolites in plasma and tissues. Indeed, the pathobiology of COVID-19 elicits substantial acute changes to the metabolome [7,8,9,10,11] that may be caused by the virus-driven mobilization of macromolecular building blocks such as nucleic acids, amino acids, and FA necessary for its replication [7,8,9,10,11,12,13,14], or by host responses to viral infection. If the latter involves mitochondrial dysfunction, the resultant incomplete FAtOx manifests as an accumulation of medium-chain acylcarnitines in plasma [15], a signature found in those with more severe acute illness or of increased age [12]. Interestingly, individuals recovering from moderate- and critical illness also exhibit higher plasma medium-chain acylcarnitines and lower levels of circulating TCA cycle intermediates such as succinate [16,17], suggesting lingering impaired mitochondrial capacity. While studies have documented prolonged alterations to the metabolome in patients who fully recovered from COVID-19 [17,18], the metabolomic signature of PASC, especially in the absence of severe acute COVID-19, has not been reported. To bridge this gap, we set out to measure metabolites in venous plasma of PASC individuals, compared with individuals who recovered from COVID-19 without PASC and healthy individuals.

## 2. Materials and Methods

### 2.1. Study Design and Human Individuals

We conducted a single-center retrospective cohort analysis, approved by the Institutional Review Board at National Jewish Health (NJH), with consent requirement waived. We identified three groups of individuals, and their informed consent-compliant de-identified blood samples were obtained from the NJH Biobank. De-identified demographic data and COVID-19 status associated with the blood samples were obtained from the bioinformatics team via NJH dataSCOUTTM. The three groups of participants included in this study (Table 1) were defined as individuals with: PASC (n = 29); a history of testing-confirmed acute COVID-19 without PASC (n = 16); and those who lacked any history of positive testing for COVID-19 (n = 30). All three groups were seen in an ambulatory clinic/capacity. Patients with COVID-19 without PASC will herein be referred to simply as post-COVID. All participants enrolled were of at least 18 years of age and had blood samples in the Biobank. PASC individuals were patients evaluated at the NJH Center for Post-COVID Care between 16 March 2020, and 22 March 2020. They were enrolled only if blood was collected > 28 days after testing positive by SARS-CoV-2 PCR and were experiencing fatigue or exercise intolerance, consistent with PASC. Exclusion criteria were: (a) abnormal CT chest findings due to COVID-19, (b) baseline cardiomyopathy defined as ejection fraction (EF) < 50%, (c) chronic severe pulmonary disease and/or impaired baseline oxygen saturation (SpO_2_) and/or chronic hypoxia requiring supplemental O_2_, (d) chronic myopathy for any reason, (e) post-ICU admission within 12 months, and (f) acute COVID-19 infection requiring hospitalization. De-identified blood samples were transferred to the University of Colorado School of Medicine Metabolomics Core for processing and metabolomic analysis, under a mutually approved Material Transfer Agreement. Further clinical data, such as retrospective baseline laboratory values, radiography, echocardiography, and pulmonary function testing, were obtained in a de-identified manner via the NJH Bioinformatics team.

### 2.2. Sample Processing and Metabolite Extraction

Plasma samples were extracted as described [19], by adding 240 μL of ice-cold methanol/acetonitrile/water 5:3:2 to 10 μL of plasma. Samples were then vortexed at 4 °C for 30 min and centrifuged at 14,000 × *g* for 10 min at 4 °C. Supernatants were transferred to a new tube for metabolomics analysis.

### 2.3. Ultra-High-Pressure Liquid Chromatography (UHPLC)-Mass Spectrometry (MS) Metabolomics and Lipidomics

Analyses were performed using a Vanquish UHPLC coupled online to a Q Exactive mass spectrometer (Thermo Fisher, Bremen, Germany). Samples were analyzed using a 5 min gradient, as described [20,21]. Solvents were supplemented with 0.1% formic acid for positive mode runs and 1 mM ammonium acetate for negative mode runs. MS acquisition, data analysis, and elaboration were performed as described [20,21].

### 2.4. Metabolomics

UHPLC-MS metabolomics analyses were performed as described in method 19–21 and application papers,8,22 using a Vanquish UHPLC system coupled online with a high-resolution Q Exactive MS (Thermo Fisher, Bremen, Germany). Samples were resolved over a Kinetex C18 column (2.1 × 100 mm, 1.7 µm; Phenomenex, Torrance, CA, USA) at 45 °C. A volume of 10 µL of sample extracts was injected into the UHPLC-MS. Each sample was injected and run four times with two different chromatographic and MS conditions, as follows: (1) using a 5-min gradient at 450 µL/min from 5–95% B (A: water/0.1% formic acid; B: acetonitrile/0.1% formic acid), and the MS was operated in positive mode and (2) using a 5-min gradient at 450 µL/min from 5–95% B (A: 5% acetonitrile, 95%water/1 mM ammonium acetate; B: 95%acetonitrile/5% water, 1 mM ammonium acetate) and the MS was operated in negative ion mode. The UHPLC system was coupled online with a Q Exactive (Thermo, San Jose, CA, USA) scanning in Full MS mode at 70,000 resolution in the 60–900 m/z range, 4 kV spray voltage, 15 sheath gas and 5 auxiliary gas, operated in negative or positive ion mode (separate runs).

### 2.5. Quality Control and Data Processing

Calibration was performed prior to analysis using the PierceTM Positive and Negative Ion Calibration Solutions (Thermo Fisher Scientific). Acquired data were then converted from raw to mzXML file format using Mass Matrix (Cleveland, OH, USA). Samples were analyzed in randomized order with a technical mixture (generated by mixing 5 µL of all samples tested in this study) injected every 10 runs to qualify instrument performance. This technical mixture was also injected three times per polarity mode and analyzed with the parameters above, except collision-induced dissociation (CID) fragmentation was included for unknown compound identification (10 ppm error for both positive and negative ion mode searches for intact mass, 50 ppm error tolerance for fragments in MS2 analyses).

### 2.6. Metabolite Assignment and Relative Quantitation

Metabolite assignments were performed using MAVEN (Princeton, NJ, USA) [22], against an in house library of 3000 unlabeled (MSMLS, IROATech, Bolton, MA, USA; IroaTech; product A2574 by ApexBio; standard compounds for central carbon and nitrogen pathways from SIGMA Aldrich, St Louis, MO, USA).

### 2.7. Statistics

Demographic and clinical characteristics were compared between the three study groups. Metabolomic data were normalized in MetaboAnalyst 5.0 [23], via Autoscale normalization (i.e., mean-centered and divided by the standard deviation of each variable). Global differences in mean metabolic values between the three study groups were assessed using one-way ANOVA. Pairwise differences in metabolic values between PASC, post-COVID, and healthy control groups were carried out using two-sample *t*-tests assuming equal variance. Benjamini–Hochberg (BH) adjustments to p-values were implemented to control for a false discovery rate (FDR) ≤0.05. Metabolic profiles were assessed using partial-least squares discriminant analysis (PLS-DA) and important metabolites were identified by a variable importance in projection (VIP) score >1.5. Additionally, heat maps of the top 25 significant metabolites (via ANOVA) were created. Volcano plots of log-transformed p-values and log-transformed fold-change for all pairwise comparisons of metabolite values were created. Differences in reported PASC symptoms between PASC and post-COVID groups were assessed using Chi-squared or Fisher’s exact tests. Statistical significance was considered at the α = 0.05 significance level. Analyses were carried out in GraphPad Prism software and R software version 4.1.2.

## 3. Results

### 3.1. Characteristics of the Participants

From the initial 76 individual samples obtained for this study, one sample (from the PASC group) was excluded from analysis due to outlier data indicating technical errors in collection, storage or processing. Of the 75 participants included in the analysis (Table 1), 59% identified as female, 83% as non-Hispanic, and 77% as White. Individuals in the PASC group were younger (42 years vs. 60 years in post-COVID, *p* < 0.05). Median body mass indices (BMI) were > 25 kg/m^2^ and did not differ between PASC and post-COVID individuals. Fewer than half of individuals had history of pulmonary diagnoses, with asthma being the most common (28% of PASC and 44% of post-COVID), and most were never-smokers. Consistent with the exclusion criteria, all included participants lacked any radiographic changes of acute lung disease. Diagnostic testing and medication data were inconsistently available in the three studied groups, particularly amongst healthy controls. No significant differences noted between PASC and post-COVID individuals in the available laboratory values were deemed potentially relevant, such as hemoglobin, transaminases (AST, ALT), creatinine, glucose, bilirubin, albumin, or alkaline phosphatase, consistent with previous reports [23]. Spirometry data were within normal limits and similar between the two groups (mean FEV_1_ %predicted of 103% and 92% in PASC and post-COVID, respectively, *p* = 0.12). Of the few individuals with available echocardiography results (8 PASC and 5 post-COVID), all showed normal findings (e.g., left ventricular EF > 60%). SpO_2_ values were normal in both PASC and post-COVID groups. Interestingly, post-COVID individuals had a higher use of inhaled and systemic corticosteroids or immunosuppressant medications compared to PASC individuals (Table 1).

The most common symptoms reported in the PASC and post-COVID groups are listed in Table 2. As anticipated, PASC individuals had higher frequency than post-COVID individuals of dyspnea (15 vs. 3) and brain fog (9 vs. 0) (*p* = 0.031 by Chi-squared test and *p* = 0.017 by Fisher’s exact test, respectively). Other common complaints amongst PASC were fatigue, palpitations and chest discomfort.

### 3.2. Plasma Metabolic Phenotypes in PASC Compared to Those Fully Recovered from COVID-19 (Post-COVID) and Healthy Controls

Metabolomics analyses were performed on plasma of individuals from PASC (n = 29), COVID-19 fully recovered without PASC (n = 16), and healthy controls (n = 30) (Figure 1A). Partial least square-discriminant analysis (PLS-DA) of the metabolomics data (Appendix A) showed that the three groups separated across principal component 1 (PC1), which explained 12.1% of the total variance (Figure 1B). Variable importance in projection (VIP) analysis of the top 15 metabolites with the highest loading weight on PC1 (Figure 1C) ranked lactate and pyruvate highest, followed by long-chain acyl-carnitines and free fatty acids, and the carboxylic acids 2-hydroxyglutarate and malate. Pathway analysis of the PASC plasma features that were statistically different from the other groups (by ANOVA) revealed marked alterations in the fatty acid biosynthesis/activation/metabolism/beta oxidation and the TCA cycle pathways (Figure 1D). Many of these compounds were also identified in the top 25 metabolites that were most significantly changed (by ANOVA) (Figure 1E).

**Figure 1 metabolites-12-01026-f001:**
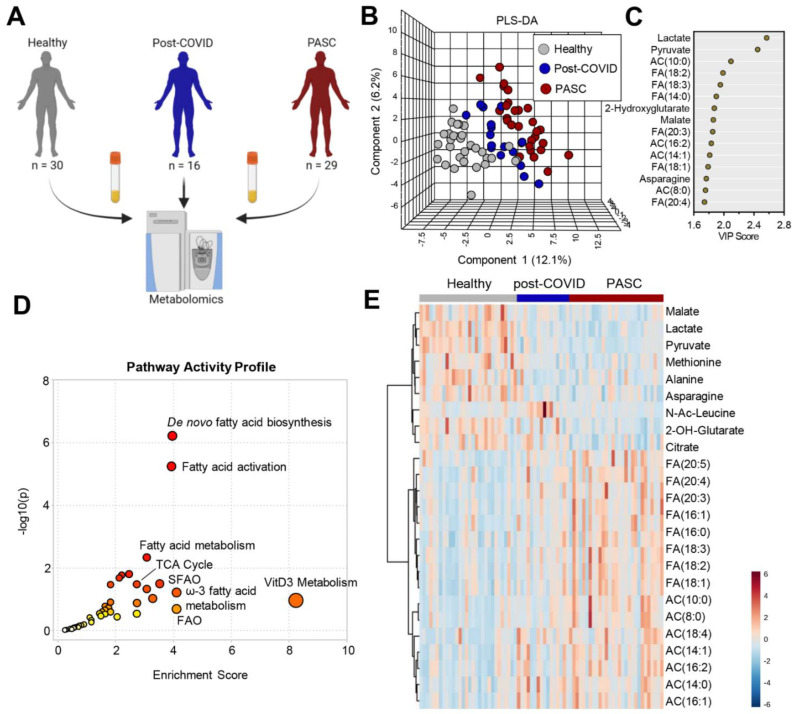
Metabolomics profiles of PASC Plasma. (**A**) Plasma from healthy (n = 30), post-COVID without PASC (n = 16), and PASC (n = 29) individuals was collected at rest and assessed by nontargeted metabolomics using LC-MS. (**B**) Partial Least-Squares Discriminant Analysis (PLS-DA) of metabolomics data. (**C**) Variable Importance in Projection (VIP) scores of the top 15 metabolites contributing to PLS-DA clustering pattern. (**D**) Enriched metabolic pathways of annotated untargeted negative polarity LC-MS features. The size of each circle corresponds to its enrichment factor and color corresponds to p-value (from white to red). Fatty acid oxidation (FAO), saturated fatty acid oxidation (SFAO). (**E**) Heat map with top 25 metabolites significantly different among the 3 groups of individuals based on ANOVA (red color indicates higher levels and blue indicates lower levels in pairwise comparisons).Separate comparisons of plasma between healthy individuals and post-COVID (Figure 2A) or PASC (Figure 2B) revealed distinct and marked metabolic phenotypes between the groups. A direct comparison between post-COVID and PASC individuals by PLS-DA highlights that these two groups can be only subtly distinguished by untargeted metabolomics of plasma at rest (Figure 2C), with 7 significantly lower metabolites and 1 significantly higher metabolite in PASC plasma (Figure 2D).

**Figure 2 metabolites-12-01026-f002:**
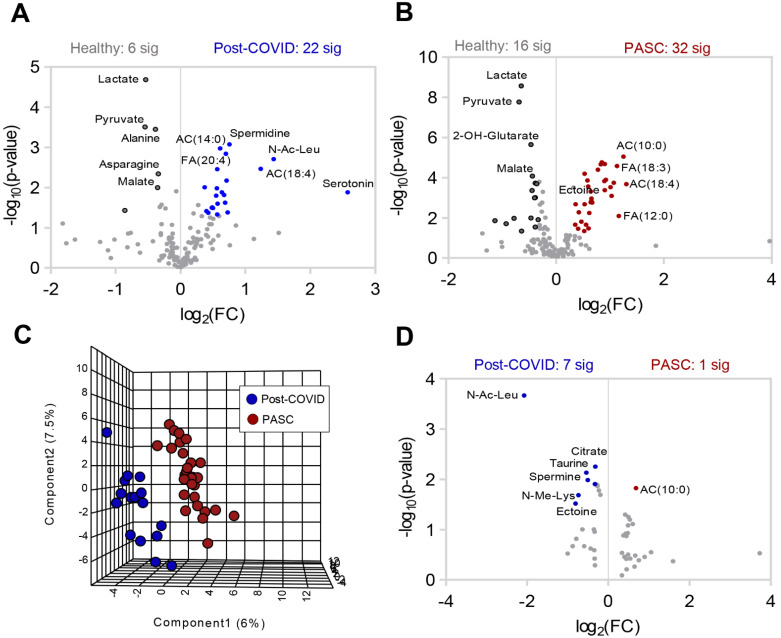
Differences in plasma metabolite profiles in individual groups. Volcano plot comparisons to healthy plasma of (**A**) post-COVID without PASC and (**B**) PASC. C-D Comparison of post COVID without PASC vs. PASC shown by (**C**) PLS-DA and (**D**) volcano plot.

### 3.3. Higher Levels of Fatty Acid Metabolites in PASC Plasma

Compared to the other two groups, PASC plasma exhibited significantly higher acylcarnitines and free fatty acids (Figure 3A,B, respectively). These included saturated (C8, C10, C14, C16), monounsaturated (C14:1, C16:1), poly-unsaturated (16:2, 18:3), and hydroxylated (C12-OH, C14-OH, C16-OH, C18-OH, C14:1-OH, C16:1-OH) acylcarnitines (Figure 3A); as well as saturated (C10, C12, C14, C16), monounsaturated (C14:1, C16:1, C18:1), and poly- and highly unsaturated very-long-chain (C18:2, 18:3, 20:3, 20:4, 20:5 and 22:6) free fatty acids (Figure 3B). BH FDR adjustment for all metabolites are reported in Appendix A. The most significant changes among all three groups were noted in acylcarnitine 10:0, 14:1, 16:2, and 14:0 and fatty acids 18:2, 1:3, 14:0, 20:3, and 18:1.

### 3.4. Lower Levels of Mono-, Di- and Tri-Carboxylates in PASC Plasma

Compared to the other two groups, PASC plasma exhibited lower levels of in mono- (pyruvate, lactate), di- (succinate, malate, 2-hydroxyglutarate) and tri-carboxylates (citrate) (Figure 4).

### 3.5. Lower Levels of Amino Acid Metabolites in PASC Plasma

Amino acids tended to be lower in both PASC and post-COVID individuals, with markedly lower levels in PASC (Figure 5). While both groups had lower levels of alanine, asparagine, methionine, and threonine, PASC plasma additionally had lower levels of leucine/isoleucine, proline, tryptophan, tyrosine, and valine. In addition, PASC levels of tyrosine were lower than those of post-COVID individuals.

## 4. Discussion

These results, which, to our knowledge, provide the first characterization of the plasma metabolome in individuals with PASC, indicate several major metabolic derangements. In the context of previously published metabolomic studies of individuals with exercise intolerance resulting from other etiologies and interpreted in light of the clinical manifestations of PASC, our data provide a novel insight into the potential role of metabolism in the pathogenesis of this condition. Although the metabolic pathways associated with prognosis and severity of COVID-19 have been studied [24,25,26,27], and recent studies have documented the lingering metabolic effects of COVID-19, even in patients who fully recovered [17,18], little is known about development of PASC and related exercise intolerance. The risk factors and mechanisms of PASC development are poorly understood, despite its affecting a large number of individuals: ~60% of COVID-19 complicated by severe pneumonia and 37% of COVID-19 patients without pneumonia [28]. Given the persistence of fatigue and exercise intolerance of more than 6 months in more than half of COVID-19 survivors [29], there is a clear need to understand and manage PASC. Demographically, our PASC cohort was similar to others that have been reported, indicating that individuals with PASC are young and without pre-morbid pulmonary functional impairment [30]. As with other autoimmune and inflammatory conditions that manifest with increased severity in younger age groups, we noted that demographic differences in our groups, with PASC individuals being younger than post-COVID without PASC. In addition to the potential impact of an exuberant immune response, this difference may also be due to our study design, which captured individuals who presented for medical care. Younger individuals who develop PASC may seek medical attention sooner than those of older age, who may be less concerned or attribute PASC symptoms to other comorbidities or advanced age. We recently reported that patients who developed PASC with exercise intolerance, despite mild acute COVID-19, exhibit lactate accumulation and respiratory gas exchange indicative of impaired fatty acid oxidation during exercise challenge, suggesting mitochondrial dysfunction [2]. The results of the plasma metabolomics obtained during rest in the current study are consistent with this report and support the hypothesis that a dysfunction in substrate utilization in mitochondria underlies the metabolic manifestations of PASC.

Although our study is of limited size and did not follow individuals longitudinally across the disease spectrum, by using an observational design that included both individuals who fully recovered from COVID-19 and those with PASC, our data provide insight into the temporal evolution of metabolic derangements in COVID-19. Our group was among the first to provide evidence of metabolic dysregulation in patients with acute COVID-19 as a function of disease severity correlated to circulating levels of inflammatory cytokines such as IL-6 [11,12]. When compared to those abnormalities, our results indicate that several metabolomic abnormalities discovered in acute COVID-19 may be persistent in those with PASC. Metabolomic differences and particularly fatty acid pathways identified in our ambulatory cohort of PASC individuals are concordant with findings of others investigating the severity of acute hospitalized COVID-19 patients [31].

The higher levels of plasma carnitine-conjugated and free fatty acids, especially poly- and highly unsaturated, as well as hydroxylated, fatty acids in PASC are consistent with the lower fatty acid oxidation capacity of mitochondria. This metabolomic signature was reported in acute COVID-19, where disease severity was associated with both dyslipidemia and markers of mitochondrial dysfunction [32]. Whereas the impact of higher levels of circulating free fatty acids on functional manifestations of PASC is unclear, during acute disease, they are suspected to promote and sustain viral particle formation [12]. Of note, the accumulation of specific lipid classes in plasma (especially free and carnitine-conjugated fatty acids) was associated with erythrocyte dysfunction 8, which would, in turn, impair oxygen delivery to target organs and, therefore, decrease the oxidative processes necessary for fatty acid substrate utilization in mitochondria. Since red blood cells’ lifespan is ~120 days, these abnormalities may persist for months after COVID-19 and could explain PASC symptoms such as fatigue and exercise intolerance. There are several lines of evidence that oxygen consumption is impaired in COVID-19 survivors, as measured by VO_2_ kinetics shown during on-time 85% oxygen deficit, 28% greater mean response time in this population, with an 11% longer half-time of VO_2_ recovery of at the off-transient [33]. Indeed, VO_2_ max was impaired (albeit modestly) in a cohort of 50 individuals with PASC studied during graded maximal exercise testing, that also exhibited indirect evidence of markedly reduced FatOx capacity [2]. Although in that cohort, the CHOx capacity was slightly reduced, it was not significantly different from historic cohorts of healthy individuals. In the current study of PASC plasma obtained in resting individuals, compared to controls, we found significantly lower circulating levels of carboxylic acids (mono-, di- and tri-carboxylates). These changes are suggestive of impaired pyruvate/lactate metabolism, which may occur at the level of mitochondrial catabolism. Long considered a “dead-end” metabolic product of anaerobic metabolism (glycolysis) [34], lactate is an important substrate for oxidative metabolism, gluconeogenesis, muscle glycogenesis, and a regulator of FAtOx [35,36,37].

Interestingly, the PASC metabolomic phenotypes of higher levels of circulating carnitines and lower levels of carboxylic acids are similar to those noted in individuals with sickle cell traits [38] who carry a single copy of genetically altered beta globin gene (E6V). These individuals, while phenotypically silent (asymptomatic) at rest, are exercise or high-altitude intolerant. Interestingly, COVID-19 causes metabolic, structural and morphological alterations to the erythrocytes [10,39,40], which may circulate for up to 120 days, well past the clearance of the infection, and yet contribute to dysregulated oxygen kinetics for up to 4 months post-viral-exposure [41]. Moreover, higher baseline levels of fatty acids and acylcarnitines (including marked increases in medium chain lengths) are indicative of ongoing mobilization of fatty acids but show impaired ability for oxidation due to mitochondrial dysfunction. Importantly, these signatures have also been observed in patients with Type 2 diabetes [42,43,44,45,46] and sepsis [47,48,49,50,51], both pathologies that stem from metabolic and mitochondrial dysfunction. These signatures are recapitulated in amateur [52] and elite cyclists at exhaustion and are a marker of exertion [53,54]. Thus, it is relevant that incomplete fatty acid oxidation presents in PASC patients at baseline, and this rationalizes future studies to determine whether the PASC phenotypes we reported here are modulated by moderate/intense exercise, and whether specific interventions can be envisaged to restore fat-burning capacity. In the latter case, targets may include both the normalization of oxygen-carrying/delivery capacity by circulating erythrocytes, or by molecular or training strategies (e.g., long aerobic training [55,56]) aimed at increasing mitochondrial mass and/or restoring optimal mitochondrial function.

We also identified that PASC plasma was characterized by a significant depletion of multiple amino acids, including those (alanine, aspartate, asparagine, and serine) that are involved in transamination reactions. The depletion of methionine may reflect higher levels of oxidant stress-induced isoaspartyl-damage repair, as reported in red blood cells of COVID-19 patients [10], or alternative methionine use, for example to fuel long-term epigenetic regulation (e.g., DNA methylation at CpG island; methyl-6-adenosine RNA modification, histone methylation). The lower levels of branched-chain amino acids leucine/isoleucine and valine is consistent with altered catabolism, perhaps as compensatory mechanism to overcome blockade in fatty acid oxidation. Alternatively, alterations in dietary intake and/or exercise regimens may drive the depletion in the levels of these amino acids in this population. It is worth noting that the depletion of taurine is a hallmark of COVID-19 disease severity [11], to the extent that taurine supplementation has been proposed as an adjuvant in the treatment of COVID-19 [57]. Here, we show that circulating levels of taurine are restored to healthy control levels in post-COVID patients but not in PASC. Of note, depletion in tryptophan is consistent with observations in acute COVID-19 patients [8,11,12,58,59,60]. During acute stages of COVID-19, we linked the tryptophan-kynurenine metabolism to the activation of antiviral responses, seroconversion and resolution of the infection, and overall disease severity and prognosis15. The activation of the cGAS/STING/interferon signaling axis upregulates tryptophan consumption to fuel the immunomodulatory [32] kynurenine pathway in a sex-, age- and body-mass-index-specific manner. Interestingly, the lower levels of tyrosine and related catabolites at baseline in PASC relative to healthy plasma mirror a relationship between elite cyclists stratified by endurance capacity that is dependent upon mitochondrial function [53], further suggesting a link between these amino acids and exercise tolerance.

We recognize that this study holds several limitations, including the small sample size, which precluded adequate stratification of metabolomic signatures based on prior COVID-19 disease severity, correlation with specific PASC symptoms, or other biological factors such as sex, age, or ethnicity. Owing to the sample size, the study was not sufficiently powered to afford correction for the impact of such variables on the metabolome of patients with PASC, a limitation that will be addressed in currently ongoing follow-up studies. Samples tested in this study were collected at rest, while a more clearly dysfunctional metabolic signature would likely manifest itself upon exercise challenge. In addition, only steady-state plasma analyses were performed, which clearly highlight a signature of dysfunctional fatty acid catabolism and also highlight the need to determine sources of incompletely oxidized fatty acids. Nevertheless, our data offer compelling evidence of metabolic dysfunction in PASC and should fuel future investigations of oxygen and lactate kinetics and mitochondria biology.

In conclusion, our novel findings offer compelling evidence of metabolic dysfunction in PASC and should fuel future investigations of oxygen and lactate kinetics and mitochondria biology. Furthermore, the appreciation of the long-lasting derangement in fatty acid catabolism in PASC could pave the way for interventions aimed at promoting mitochondrial biogenesis and restoring fat-oxidation capacity. Although such interventions and strategies in PASC are awaiting further research, low-intensity or interval exercise regimens have been shown to exert such an effect by boosting metabolic flexibility in healthy athletes and less fit individuals [55,61,62], including sedentary individuals [63], and patients with chronic obstructive pulmonary disease [64,65].

## Figures and Tables

**Figure 3 metabolites-12-01026-f003:**
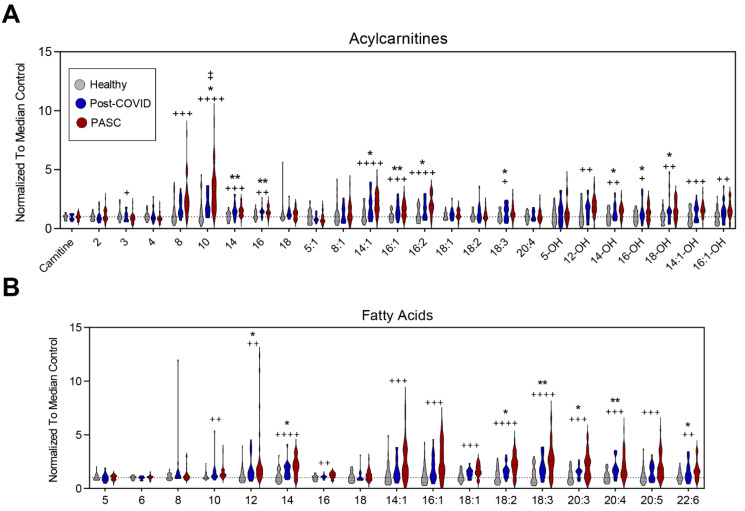
Fatty Acid Oxidation. (**A**) Relative levels of acylcarnitines and (**B**) free fatty acids are depicted as violin plots normalized to median control value. Statistically significant changes noted between * controls and post-COVID, + control and PASC, and ‡ post-COVID and PASC; FDR uncorrected * or + *p* < 0.05; ** or ++ *p* < 0.01, +++ *p* < 0.001, ++++ *p* < 0.0001.

**Figure 4 metabolites-12-01026-f004:**
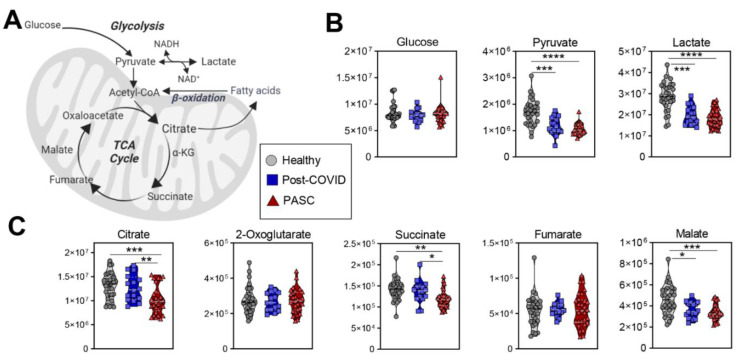
Glycolysis and TCA Cycle. (**A**) A pathway map for energy metabolism is shown, along with metabolite values for (**B**) glycolysis, and (**C**) the tricarboxylic acid (TCA) cycle. FDR uncorrected * *p* < 0.05; ** *p* < 0.01, *** *p* < 0.001, **** *p* < 0.0001.

**Figure 5 metabolites-12-01026-f005:**
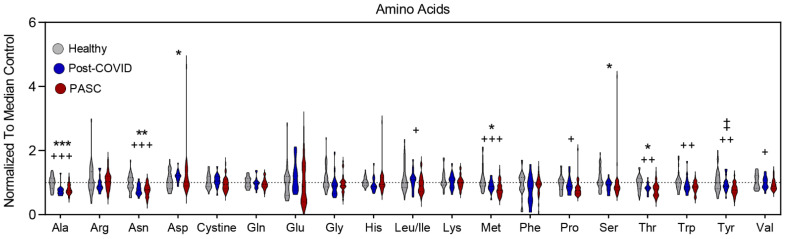
Changes in Amino Acids profiles. The relative levels of amino acids reported by single letter code are depicted as violin plots normalized to median control value. To identify significant comparisons, * healthy and COVID19 without PASC, + healthy and PASC, and ‡ COVID and PASC with FDR uncorrected * or + *p* < 0.05; ** or ++ *p* < 0.01, *** or +++ *p* < 0.001.

**Table 1 metabolites-12-01026-t001:** Individual characteristics by COVID-19 status (from 16 March 2020 to 22 March 2020).

Characteristic	COVID (+)	COVID (+)	COVID (-)
	PASC	Post-COVID	Controls
	(n = 29)	(n = 16)	(n = 30)
Age (years), mean ± SD	42 ± 13	60 ± 14	48 ± 11
Male/Female, n	12/17	8/8	19/11
BMI (kg/m^2^), mean ± SD	27.5 ± 7	25.9 ± 3.9	N/A
Race, n (% total)			
White	22 (76%)	12 (75%)	24 (80%)
Black or African American	0	0	0
American Indian/Alaska Native	0	0	0
Native Hawaiian/Pacific Islander	0	0	0
Asian	2 (7%)	0	1 (3%)
Unknown or declined or multiple	5 (17%)	4 (25%)	5 (17%)
Ethnicity, n (% total)			
Hispanic or Latino	1 (3%)	4 (25%)	2 (7%)
Non-Hispanic	26 (90%)	12 (75%)	24 (80%)
Unknown or declined	2 (7%)	0	4 (13%)
Smoking status, n (% total)	29 (100%)	11 (69%)	N/A
Ever smoker	8 (28%)	3 (27%)	
Never smoker	21 (72%)	8 (73%)	
COPD, n (% total)	0	0	N/A
Asthma, n (% total)	8 (28%)	7 (44%)	N/A
Chronic heart disease (includes arrhythmias)	4 (14%)	2 (12%)	N/A
Diabetes, n (% total)	4 (14%)	3 (19%)	N/A
Hyperlipidemia, n (% total)	1 (3%)	2 (12%)	N/A
Medications, n (% total)			N/A
Corticosteroids/ immunosuppressants	1 (3%)	5 (31%)	
Inhaled corticosteroids	6 (21%)	7 (44%)	
Insulin	3 (10%)	0	
Anti-hyperlipidemic agents	3 (10%)	2 (12%)	
Pulmonary function test, n (% total)	17 (59%)	8 (50%)	N/A
FEV_1_ pre-bronchodilator, % predicted	103%	92%	
FEV_1_/FVC pre-bronchodilator	0.81	0.77	
Laboratory tests, n PASC, n Post-COVID			N/A
CRP (mg/dL), n = 23, n = 3	0.38 ± 0.58	0.08 ± 0.07	
Hb (g/dL), n = 26, n = 7	14.9 ± 1.4	15 ± 1.4	
ALT (U/L), n = 23, n = 6	25 ± 14	22.5 ± 11.5	
Albumin (g/dL), n = 23, n = 6	4.5 ± 0.39	4.6 ± 0.3	
Alkaline phosphatase (U/L), n = 23	65 ± 18	62.7 ± 14	
AST (U/L), n = 25	20 ± 8	18.8 ± 2.6	
Bilirubin (mg/dL), n = 25	0.99 ± 1.24	0.77 ± 0.2	
Creatinine (mg/dL), n = 25	0.9 ± 0.16	0.95 ± 0.12	
SPO_2_ awake at rest (%), n = 8, n = 2	97 ± 1.9	96.5 ± 0.7	
LVEF% biplane, n = 8, n = 5	60 ± 4	62.7 ± 5.3	N/A

**Table 2 metabolites-12-01026-t002:** Symptoms Assessed Post-COVID.

PASC Associated Symptoms	COVID (+)PASC (n = 29)	COVID (+)No PASC (n = 16)
Fatigue	10 (34%)	1 (6%)
Dyspnea	15 (52%)	3 (19%)
Exercise intolerance	1 (3%)	0
Cough	3 (10%)	4 (25%)
Fever	0	0
Myalgia	1 (3%)	0
Chest discomfort	5 (17%)	0
Headache	4 (14%)	0
Brain fog	9 (31%)	0
Diarrhea	0	0
Nasal congestion	0	0
Anosmia	1 (3%)	2 (13%)
Dysgeusia	0	0
Nausea	0	0
Abdominal pain	0	0
Vomiting	0	0
Blood clot	0	0
Palpitations	6 (21%)	1 (6%)

## Data Availability

All the raw data elaborated in this study are provided in Appendix A.

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
