# Peer review of "Signatures of Mitochondrial Dysfunction and Impaired Fatty Acid Metabolism in Plasma of Patients with Post-Acute Sequelae of COVID-19 (PASC)"

_metabolites, 2022, doi:10.3390/metabo12111026_

Round 1

Reviewer 1 Report

The manuscript (ID: metabolites-1993365) “Signatures of Mitochondrial Dysfunction and Impaired Fatty Acid Metabolism in Plasma of Patients with Post-Acute Sequelae of COVID-19 (PASC)” explores the different metabolic signature of patients that developed the post covid 19 syndrome.

As the author states in the discussion, the manuscript has several limitations and a longitudinal study as well as investigation of the metabolic changes due to physical exercise would strengthen such a study.

However, I believe this prelimiary dataset is of interest and provide initial information of the metabolic dysfunction due to PASC.

I would suggest some changes to the authors:

Several parameters related to clinics and ethnicity and risk factors have been recorded for the two cohorts PASC and Post Covid. I was wondering if some statistical approach has been used to correct for risk factors, ethnicity and clinical parameters.

It is not clear when false discovery rate or correction for multiple testing have been used. In material and methods, it is stated that more detailed can be found in figure legends, but this reviewer was not able to find any information in figure legends. Please insert in material and methods all details regarding statistical analysis.

There is a typo in figure 5 legends, mentioning table 1. Please check all manuscript for typos.

Author Response

Detailed Responses to the Reviewers’ comments

REVIEWER 1: The manuscript (ID: metabolites-1993365) “Signatures of Mitochondrial Dysfunction and Impaired Fatty Acid Metabolism in Plasma of Patients with Post-Acute Sequelae of COVID-19 (PASC)” explores the different metabolic signature of patients that developed the post COVID 19 syndrome. As the author states in the discussion, the manuscript has several limitations and a longitudinal study as well as investigation of the metabolic changes due to physical exercise would strengthen such a study. However, I believe this preliminary dataset is of interest and provide initial information of the metabolic dysfunction due to PASC.

Authors’ reply: Thank you for the extremely kind and constructive feedback. All the requested changes have been implemented in the revised version of the manuscript.

Reviewer’ comment: I would suggest some changes to the authors:

Several parameters related to clinics and ethnicity and risk factors have been recorded for the two cohorts PASC and Post Covid. I was wondering if some statistical approach has been used to correct for risk factors, ethnicity and clinical parameters. It is not clear when false discovery rate or correction for multiple testing have been used. In material and methods, it is stated that more detailed can be found in figure legends, but this reviewer was not able to find any information in figure legends. Please insert in material and methods all details regarding statistical analysis.

There is a typo in figure 5 legends, mentioning table 1. Please check all manuscript for typos.

Authors’ reply: Thank you for the constructive feedback. As noted by Reviewer 2 as well, the limited size of the cohort enrolled in this study precluded any correction based on subject sex and age, ethnicity or other potential confounders. While well aware of the role of these biological factors on COVID-19 disease severity and PASC (see comments on PASC incidence and sex within the manuscript), we were not powered enough to perform such correction herein. This is now clearly stated in the manuscript. On the other hand, all metabolomics results were FDR corrected in Metaboanalyst, as part of the standard workflow through that software. Indeed, Supplementary table 1 already includes a sheet with p-values, FDR corrected values and ranking across the three groups. All these changes are now highlighted in yellow. This is now stated as well in the methods section.  Additional requested details regarding statistical analysis were also included in the methods section. Finally, the typo in the legend to Figure 5 has been corrected. That typo is a remnant from the Metabolites article template, which was missed during the manuscript editing phase. It has now been removed, while Table 1 is still embedded in the manuscript.

Reviewer 2 Report

Review of manuscript ID: metabolites-1993365

October 14, 2022

Guntur et al. underwent an untargeted metabolomics study to determine differences in plasma metabolites among individuals who had PASC (a.k.a. “long-COVID”), individuals who recovered from COVID, and healthy controls. The study was a cross-sectional retrospective cohort of individuals with stored plasma at the National Jewish Health Biobank. The authors found that individuals with PASC had significantly higher levels of several types of fatty acids and lower levels of amino acids and several types of tricarboxylates when compared to those who recovered from COVID and healthy controls. Individuals who recovered from COVID also differed in metabolites (e.g., those involved in fatty acid metabolism) when compared to healthy controls, though these differences were smaller when compared to differences observed between PASC individuals vs. healthy controls. The authors concluded that the metabolite differences observed among individuals at rest with PASC were similar to metabolite differences that represent mitochondrial dysfunction during exercise. The manuscript is very well-written and easy/enjoyable to read. I have only a few concerns that should be addressed.

Major concerns:

Samples: Was there a difference in the number of days from the start of COVID to the blood draw between the group that recovered from COVID versus the PASC group? Did the group that recovered from COVID also undergo phlebotomy >28 days after testing positive for COVID (similar to the PASC group)? Were there any differences in the number of days that the samples were stored at the Biobank among the three groups (i.e., were there differences in sample degradation by groups)? If these differences are present, do you think it impacted results?

Adjustments: The authors presented unadjusted associations, which given the limitations of their important study is reasonable. Though rationale on why they only report unadjusted associations should be explicitly stated in their manuscript. The authors should consider also reporting associations adjusted for at least age, in addition to the unadjusted associations currently reported. Age was extremely different among the group that recovered from COVID vs. those with PASC and the healthy controls. Therefore, it would be interesting to see how adjusting for age impacts results.

PASC and age: Can the authors add a comment after the sentence at lines 287-289 in the discussion on why they think individuals with PASC tend to be younger? What mechanism would put a younger individual at a higher risk for PASC when compared to an older individual?

Restoring mitochondrial fat burning capacity: Can the author provide some more specific examples on how an intervention can restore mitochondrial fat burning capacity among PASC individuals? Are there previous reports in different patient populations (i.e., not COVID-related research) that illustrate an improvement in mitochondrial functioning following an intervention? If this could be done then it has important implications, not only for PASC patients, but also potentially for older frail individuals as well.

Inaccurate use of longitudinal terminology: Since the study was cross-sectional, the authors should not use the words “increase” and “decrease” with regard to associations. They can replace these terms with “higher” and “lower.” When increase/decrease is used it misleads the reader into thinking that the authors conducted a longitudinal study.

Minor concerns:

- The authors may consider not using the term “subject” in their research. Some believe the term “subject” to be demeaning to the individuals who take the time to participate in research or donate their samples for future use. Consider using the term “participant” or “individual” instead.

- Table 1 could be cleaned up quite a bit. For all categorical variables in the table, the authors should add percentages, in addition to frequencies. Percentages make it easier for the reader to quickly notice differences between the groups. A few rows have repetitive information too—for example, the row that contains the total number of individuals with information on race is not needed since this is the same as the total number of individuals included in the study indicated in the very first row of the table.

- Similarly, adding percentages (in addition to the frequencies) to Table 2 would make it easier for the reader to quickly detect differences in symptoms between the two groups.

- The authors may consider adding a table to the supplemental file that includes differences on all metabolites measured in the study by groups and the corresponding p-values to refer to.

- The extra axis on Figure 2 is illegible.

- The authors mention in their discussion that their study was of an “experimental design” (lines 296-299). This is incorrect, their study was an observation design.

- The authors should consider including a conclusion paragraph at the very end of their discussion. Instead the discussion ends with a ‘limitations’ paragraph.

- Missing end of parenthesis at line 354.

Author Response

Detailed Responses to the Reviewers’ comments

REVIEWER 2: Guntur et al. underwent an untargeted metabolomics study to determine differences in plasma metabolites among individuals who had PASC (a.k.a. “long-COVID”), individuals who recovered from COVID, and healthy controls. The study was a cross-sectional retrospective cohort of individuals with stored plasma at the National Jewish Health Biobank. The authors found that individuals with PASC had significantly higher levels of several types of fatty acids and lower levels of amino acids and several types of tricarboxylates when compared to those who recovered from COVID and healthy controls. Individuals who recovered from COVID also differed in metabolites (e.g., those involved in fatty acid metabolism) when compared to healthy controls, though these differences were smaller when compared to differences observed between PASC individuals vs. healthy controls. The authors concluded that the metabolite differences observed among individuals at rest with PASC were similar to metabolite differences that represent mitochondrial dysfunction during exercise. The manuscript is very well-written and easy/enjoyable to read. I have only a few concerns that should be addressed.

Authors’ reply: Thank you for the positive and constructive feedback. All the requested changes and clarifications were introduced in the revised version of the manuscript (highlighted in yellow).

Reviewer’ comment: Samples: Was there a difference in the number of days from the start of COVID to the blood draw between the group that recovered from COVID versus the PASC group? Did the group that recovered from COVID also undergo phlebotomy >28 days after testing positive for COVID (similar to the PASC group)? Were there any differences in the number of days that the samples were stored at the Biobank among the three groups (i.e., were there differences in sample degradation by groups)? If these differences are present, do you think it impacted results?

Authors’ reply:  The PASC group was required by inclusion criteria to have blood drawn >28 days after initial COVID diagnosis. Based on well-accepted definitions, >28 days is defined as prolonged symptom course in PASC. The COVID group that recovered from initial symptoms, blood draw occurred from 5 days to 17 months after initial COVID diagnosis. This group was determined by clinical evaluation to lack did COVID-related symptoms at the time of blood draw.  Blood samples from the healthy control group were collected prior to blood samples from the COVID groups. No sample degradation would be expected between the 3 groups, as they were all stored in -80ºC.  Samples are stored in aliquots and all samples were extracted from storage in one batch, which would eliminate batch effects and risk for degradation.  Authors do not suspect that differences in sample storage would impact results.    

Reviewer’ comment: Adjustments: The authors presented unadjusted associations, which given the limitations of their important study is reasonable. Though rationale on why they only report unadjusted associations should be explicitly stated in their manuscript. The authors should consider also reporting associations adjusted for at least age, in addition to the unadjusted associations currently reported. Age was extremely different among the group that recovered from COVID vs. those with PASC and the healthy controls. Therefore, it would be interesting to see how adjusting for age impacts results.

Authors’ reply: We have revised the manuscript in the section discussing limitations, to more clearly state the rationale (small sample size) for performing unadjusted association. We are currently working on expanding our cohort of PASC participants, which should allow us to perform multiple adjusted analyses.

Reviewer’ comment: PASC and age: Can the authors add a comment after the sentence at lines 287-289 in the discussion on why they think individuals with PASC tend to be younger? What mechanism would put a younger individual at a higher risk for PASC when compared to an older individual?

Authors’ reply: As with many autoimmune and inflammatory conditions, the response to infection and injury and clinical manifestations may have gender or age determinants. The exact mechanism for demographic differences in who develops PASC has been established yet. Furthermore,  the younger age of PASC participants in our study may be linked to age-related behavioral impact of PASC symptoms prompting more individuals of younger age to seeking medical care. As requested by the reviewer, we include a comment reflecting these considerations in the revised manuscript

Reviewer’ comment: Restoring mitochondrial fat burning capacity: Can the author provide some more specific examples on how an intervention can restore mitochondrial fat burning capacity among PASC individuals? Are there previous reports in different patient populations (i.e., not COVID-related research) that illustrate an improvement in mitochondrial functioning following an intervention? If this could be done then it has important implications, not only for PASC patients, but also potentially for older frail individuals as well.

Authors’ reply: We included in the conclusion a statement to reflect potential interventions that have been shown to improve mitochondrial function in other cohorts.

Reviewer’ comment: Inaccurate use of longitudinal terminology: Since the study was cross-sectional, the authors should not use the words “increase” and “decrease” with regard to associations. They can replace these terms with “higher” and “lower.” When increase/decrease is used it misleads the reader into thinking that the authors conducted a longitudinal study.

Authors’ reply: We agree with the Reviewer that our study was not longitudinal. Indeed, in the discussion section we clearly state that “Although our study is of limited size and did not follow individuals longitudinally across the disease spectrum, …“. Increase or decrease were replaced by higher and lower. Thank you for the important point.

Reviewer’ comment: - The authors may consider not using the term “subject” in their research. Some believe the term “subject” to be demeaning to the individuals who take the time to participate in research or donate their samples for future use. Consider using the term “participant” or “individual” instead.

Authors’ reply: Revised throughout.

Reviewer’ comment: - Table 1 could be cleaned up quite a bit. For all categorical variables in the table, the authors should add percentages, in addition to frequencies. Percentages make it easier for the reader to quickly notice differences between the groups. A few rows have repetitive information too—for example, the row that contains the total number of individuals with information on race is not needed since this is the same as the total number of individuals included in the study indicated in the very first row of the table.

Authors’ reply: Revised as suggested.

Reviewer’ comment: - Similarly, adding percentages (in addition to the frequencies) to Table 2 would make it easier for the reader to quickly detect differences in symptoms between the two groups.

Authors’ reply: Revised as suggested.

Reviewer’ comment: - The authors may consider adding a table to the supplemental file that includes differences on all metabolites measured in the study by groups and the corresponding p-values to refer to.

Authors’ reply: Supplementary table 1 already includes a sheet with p-values, FDR corrected values and ranking across the three groups.

Reviewer’ comment: - The extra axis on Figure 2 is illegible.

Authors’ reply: We made sure that the y axis of panel C (smallest one) in Figure 2 was readable in the printed version of the article.

Reviewer’ comment: - The authors mention in their discussion that their study was of an “experimental design” (lines 296-299). This is incorrect, their study was an observation design.

Authors’ reply: Revised as suggested.

Reviewer’ comment: - The authors should consider including a conclusion paragraph at the very end of their discussion. Instead the discussion ends with a ‘limitations’ paragraph.

Authors’ reply: Revised as suggested.

Reviewer’ comment: - Missing end of parenthesis at line 354.

Authors’ reply: Revised as suggested.